Isolation and identification of serotonin compound from banana hump: a reproductive stimulant for tiger shrimp Penaeus monodon broodstock enhancement

Parenrengi Andi 1 andi053@brin.go.id
Suryati Emma 1
Syah Rachman 1
Tenriulo Andi 1
Lante Samuel 1
Zainuddin Elmi Nurhaidah 2 elmi18id@gmail.com
Aliah Ratu Siti 1
Farizah Nuril 1
Nawang Agus 1
Sulaeman Sulaeman 1
Makmur Makmur 1
Rosmiati Rosmiati 1
Gunarto Gunarto 1
Herlinah Herlinah 1
1 Research Center for Fishery, National Research and Innovation Agency , Cibinong , Indonesia
2 Faculty of Marine Science and Fisheries, Universitas Hasanuddin , Makassar , Indonesia
Beddoe Travis
Electronic publication date: 2024 Dec 13
Publication date: 2024
Volume: 12
Electronic Location ID: e18670
Received 2024 Sep 4; Accepted 2024 Nov 19
Copyright: © 2024 Parenrengi et al.
Copyright year: 2024
Copyright holder: Parenrengi et al.
License: This is an open access article distributed under the terms of the Creative Commons Attribution License, which permits unrestricted use, distribution, reproduction and adaptation in any medium and for any purpose provided that it is properly attributed. For attribution, the original author(s), title, publication source (PeerJ) and either DOI or URL of the article must be cited.
License URL: https://creativecommons.org/licenses/by/4.0/

Keywords: Serotonin, Banana hump, Reproductive performance, Tiger shrimp, Penaeus monodon, Broodstock enhancement, Molting, Gene expression, Domestication, Chemical structure analysis

Funding: Ministry of Finance of Indonesia RIIM Program Phase-2 through the National Research and Innovation Agency (BRIN) 82/II.7/HK/2022 The Laboratory Work was Carried Out at BRIN Research Institute for Brackishwater Aquaculture and Fisheries Extension Universitas Hasanuddin This study was supported by the Ministry of Finance of Indonesia, the RIIM Program Phase-2 through the National Research and Innovation Agency (BRIN) in the grand no: 82/II.7/HK/2022. The laboratory work was carried out at BRIN, Research Institute for Brackishwater Aquaculture and Fisheries Extension, and Universitas Hasanuddin. The funders had no role in study design, data collection and analysis, decision to publish, or preparation of the manuscript.

==============================
Background

The banana plant is claimed to contain a serotonin compound that has the potential to stimulate and improve the reproductive performance of crustacean species. This study aimed to isolate and characterize the serotonin compound from the banana hump and its application to enhance the reproductive performance of tiger shrimp broodstock.

Methods

Banana hump as a part of the plant was extracted by using the maceration technique. The chemical structure of the serotonin compound was identified and characterized based on spectroscopic data, including high-performance liquid chromatography (HPLC), ultraviolet-visible (UV-Vis) spectroscopy, and infrared (IR) spectroscopy, and a comparison was made with the standard compound and the literature. The extract herb in a dose of 50 µg/g body weight was injected into the tiger shrimp broodstock in four-time administrations. During 2 months of broodstock gonadal maturation, the parameters of molting, reproduction, and gene expression related to reproduction were observed.

Results

Based on the chemical structure analysis, the stimulant component of the banana hump was identified as a serotonin compound (5-hydroxytryptamine) at a concentration of 0.7% of dry weight. The number of spawned broodstock was higher in the serotonin extract treatment (60%) than in the control treatment (40%), and the broodstock injected serotonin spawned up to the second re-maturation. In contrast, no re-maturation was obtained in the control treatment. The egg number was significantly higher using the serotonin extract (286,550 ± 46,402 eggs) than the control shrimp (148,585 ± 23,647 eggs), in which the serotonin extract treatment showed a comparatively larger egg diameter number. The higher expression of the genes related to female and male reproduction was observed in the tiger shrimp injected with serotonin extract than in the control treatment.

Introduction

As part of the sustainable aquaculture programs and ecologically friendly conservation efforts, tiger shrimp Penaeus monodon domestication was conducted to support the breeding strategic plan in regulating and controlling their reproductive system and ensuring their survival in pond culture. The tiger shrimp has been extensively cultivated in brackish water ponds using the larvae produced from broodstock of the wild stock population (Wong et al., 2021). To date, some studies have emphasized producing broodstock from pond culture to support the domestication process. A low gonadal maturation and mating broodstock of broodstock produced from the pond culture are still obstacles (Lante, Tenriulo & Parenrengi, 2018). The eye stalk ablation of tiger shrimp is a standard method to stimulate gonadal maturity (Aktaş & Kumlu, 2005; Tsukimura, 2001). However, this method, especially for broodstock produced from ponds, has not significantly increased reproductive performance. To produce a high-performance quality broodstock produced from the wild and culture pond for domestication purposes, some alternative strategies have been assessed by using external induction herb extract (Rosmiati, Lante & Suryati, 2016; Saptiani, Prayitno & Anggarawati, 2021; Suryati et al., 2021). Several herbs have been reported to contain the sesquiterpene compound, which acts to enhance the performance of shrimp reproduction (Meeratana et al., 2006; Sarojini, Nagabhushanam & Fingerman, 1995; Tomy et al., 2016; Vaca & Alfaro, 2000).

Serotonin, known as 5-hydroxytryptamine (5HT), plays a role in regulating reproductive behavior and processes (the control of mating behavior, egg production, and other aspects of reproduction) and is found in many plants, including bananas, potatoes, and sedge grass (Vaca & Alfaro, 2000). Several neurohormones, such as the molt-inhibiting hormone (MIH), red pigment-dispersing hormone (RPDH), crustacean hyperglycemic hormone (CHH), and neuro-depressing hormone (NDH), are also stimulated by serotonin (Sarojini, Nagabhushanam & Fingerman, 1994). Recent advances have focused on finding more sustainable and less invasive alternatives to improve shrimp reproduction performance, such as the use of plant-derived compounds. Serotonin is known for its role as a neurohormonal regulator in crustaceans. Previous studies have demonstrated serotonin efficacy in enhancing reproductive performance in crustacean species such as Penaeus vannamei (Vaca & Alfaro, 2000), Macrobrachium rosenbergii (Meeratana et al., 2006), Procambarus clarkii (Sarojini, Nagabhushanam & Fingerman, 1994), and Penaeus indicus (Tomy et al., 2016). However, the potential of serotonin derived from natural plant sources, like banana hump, remains underexplored. This study builds on these findings by isolating serotonin from the banana hump, a readily available and sustainable source, and testing its effects on the reproductive performance of Penaeus monodon. The application of banana hump extract may be a valuable strategy to assess molting, reproduction, and gene expression related to reproduction for supporting the tiger shrimp breeding and domestication process in commercial larvae production.

The identification and characterization of the herb extract have to be conducted to ensure the compound content that will be applied to animal testing, including tiger shrimp. The serotonin compound from the banana hump needs to be validated before being injected into the tiger shrimp. This study was conducted to isolate and identify the serotonin compound of the banana hump, as well as to evaluate its effect on the molting rate, reproduction, and gene expression of tiger shrimp broodstock, compared to the control shrimp without extract herb injection.

Materials and Methods

Ethical approval

All procedures conducted in this study involving tiger shrimp (P. monodon) were approved by the Ethical Committee of the National Research and Innovation Agency (BRIN) of Indonesia with the approval certificate number: 003/KE.02/SK/01/2023. To ensure the ethical treatment of the broodstock, the injection process was closely monitored, and any potential side effects were recorded. All handling procedures followed established ethical guidelines for the use of invertebrates in research, ensuring that the well-being of the shrimp was prioritized at every stage of the experiment. Each shrimp was observed post-injection for behavioral changes, molting irregularities, or other signs of distress. In cases where broodstock exhibited signs of stress, such as reduced feeding activity, immediate corrective actions were taken. Throughout the study, no significant adverse effects were observed, confirming that the serotonin dosage and frequency were within a safe range for the broodstock.

Banana hump

The hump of the banana plant was collected from the farmer in Maros, South Sulawesi, Indonesia, and transported to the laboratory for extraction purposes. The collected banana plant was morphologically identified as a species of Musa acuminata. This species taxonomically belongs to the Class: Liliopsida, Order: Zingiberales, Family: Musaceae, Genus: Musa, and Species: Musa acuminata (Mathew & Negi, 2017). Isolation and identification of serotonin compound from banana hump were conducted at the laboratory of the Research Institute for Brackishwater Aquaculture and Fisheries Extension-Ministry of Marine Affairs and Fisheries, Universitas Hasanuddin, and National Research and Innovation Agency.

Isolation of serotonin compound

The serotonin compound from the banana hump was extracted using hot distillation water solvent, and the alcohol and acetone solvent was applied to recrystallize the substance (Erland et al., 2018). After being cleaned of dirt and sticky soil, banana humps were chopped into little pieces, dried, and then blended until smooth. The small pieces of 500 g banana hump were placed in a glass beaker and two liters of water was added, and then heated to boiling until 95 °C for 2 h. To separate the pulp, the mixture was filtered by using a sieve to get a water solution. The hot solution was divided into approximately one-seventh part, mixed with one liter of acetone at 80 °C before being combined with the remaining hot extract solution to form an easily separated precipitate. Following a night of cooling at room temperature and centrifugation, the solution was divided, the supernatant was discarded and the colorless precipitate was washed with 50% acetone. Using a rotary evaporator, the filtrate was evaporated at a temperature below 50 °C. The precipitate resulting from evaporation was dissolved with 55 mL of distilled water and heated to boil until 95 °C for 2 h. As much as 350 mL of acetone was added and stored in a cold room (20°) for 2 days. The solution was precipitated by centrifugation at 1,000 rpm, 4 °C for 2 min to produce a flesh-colored precipitate containing 200 units/mg. The filtrate was evaporated until dry, then dissolved with 50 mL of 50% methanol, evaporated until dry, then dissolved again with 10 mL of hot absolute methanol. The crystals formed were washed with acetone and methanol with decantation, which was carried out repeatedly until white crystals were obtained (Ramakrishna, Giridhar & Ravishankar, 2011).

The white crystals were purified by repeated recrystallization using acetone and alcohol until pure crystals were obtained. The purity test of serotonin obtained was carried out using high-performance liquid chromatography (HPLC) ThermoSpektro Varian. The mobile phase consisted of an 80:20 ratio of methanol to water, with a flow rate of 0.5 mL/min, and the column temperature was maintained at 25 °C. The sample injection volume was 20 µL, and the detection was carried out at a wavelength of 205 nm. The pure isolated serotonin was identified based on its melting point and elucidated its structure by using ultraviolet-visible (UV-Vis) spectroscopy and infrared (IR) spectroscopy, as well as compared to the characteristics of the serotonin standard (Ly et al., 2008).

Application of serotonin to the tiger shrimp broodstock

The serotonin extract application was conducted at the tiger shrimp hatchery in Barru, South Sulawesi, Indonesia, from February to October 2023. The tiger shrimp broodstocks (n = 20) of females (128.6 ± 23.0 g in weight and 23.2 ± 1.2 cm in length) and males (81.9 ± 19.5 g in weight and 20.4 ± 1.5 cm in length) were transported alive and acclimatized for 2 weeks at the hatchery. The shrimp was fed a mixture of squid and seaworm (50%:50%) two times daily in the morning and afternoon in a dose of 15% of body weight. The shrimp were reared in the 2 × 2 × 1 m3 concrete tanks filled with 0.5 m water depth of seawater and stocking density was 10 broodstocks per tank, consisting of five males and five females. The tanks were equipped with an aeration system, and a minimum 50% water exchange was conducted daily to maintain good water quality. Two tanks were used to apply two treatments, namely the application of serotonin extract (SE) and the application without serotonin as the control treatment (CT).

Serotonin extract was injected into the tiger shrimp broodstock once a week for 4 weeks (Rosmiati et al., 2022; Suryati et al., 2021). According to the references, the dose applied was 50 μg/g body weight (Aktaş & Kumlu, 2005; Wongprasert et al., 2006), while the control shrimp was without injection of extract herb. After four times injections, an eyestalk of female shrimp was ablated according to the method of Lante & Laining (2016). The molting, gonadal maturation level (GML), and reproduction were observed daily to determine the effect of herbal extract application. Stages of female GML are well defined for penaeid shrimp in simple scales of four stages (Budi et al., 2020; Promwikorn, Kirirat & Thaweethamsewee, 2004). They are morphologically classified as GML-1 (immature/spent stage), GML-2 (early maturing stage), GML-3 (late maturing stage), and GML-4 (the mature stage). While the determination of the male broodstock GML was carried out according to the appearance of the spermatophore around the shrimp petasma with the criteria that GML-1 is still empty (testes maturation), GML-2 is half filled (vas deferens maturation), and GML-3 is filled (spermatophore synthesis) which refers to the method developed by Alfaro (1993). For spermatozoa analysis, spermatophore was removed from male tiger shrimp broodstock using an electric shock method of 15 V, seven mA for 2–3 s (Sandifer et al., 1984).

Expression of genes related to reproduction

For analysis of gene expressions related to reproduction, as much as 100 µL of hemolymph were collected from tiger shrimp broodstocks. The RNA total extraction and cDNA synthesis were carried out based on the reference of Parenrengi et al. (2021). RNA total was isolated using a Geneaid kit and then continued with cDNA synthesis using the Ready To Go (RTG) You-Prime First Strand Beads. The cDNA was used as a DNA template in the PCR amplification process. Gene expressions of vitellogenin on female tiger shrimp and gametogenesis on male tiger shrimp were performed based on the method developed by Haryanti, Fahrudin & Sembiring (2015) and Suryati et al. (2021), using the specific primer of the vitellogenin gene (F-5′ att cgg aac gtg cat ttg ctg ca -3′ and R-5′ gtt ctc aag cat tgt gac agg att 3′) and gametogenesis gene (F-5′, agg cga cgg gtt tcg tga ca 3′ and R-5′ tga tct tct gca gtc gtg aca t 3′), and β-actin gene (F-5′ gag cga gaa atc gtt cgt gac 3′ and R-5′ gga agg aag gct gga aga gag 3′) in concentration of 1 µL of 10 ρmol stock for each. All reactions included negative controls (NC) to monitor contamination. Both vitellogenin and gametogenesis were amplified by PCR conditions of an initial denaturation at 95 °C for 15 min, followed by 40 cycles of denaturation at 95 °C for 15 s, annealing at 60 °C for 30 s, and extension at 68 °C for 1 min. The final extension was carried out at 68 °C for 2 min. The β-actin expression was carried out in the PCR condition of an initial denaturation at 95 °C for 1 min, followed by 35 cycles of denaturation at 95 °C for 15 s, annealing at 56 °C for 15 s, and extension at 72 °C for 10 s, followed by 72 °C for 5 min. To confirm the successful amplification of the target gene, PCR results were run into a 1.0% agarose gel and visualized using a gel documentation system. The thickness of the DNA band on the electrophoresis gel was an indicator of low or high levels of gene expression.

Data analysis

The HPLC, UV-Vis, and IR spectrum analysis data were determined based on the serotonin standard compound and descriptively discussed. The molting, GML, egg number and egg diameter, spermatophore weight, number, and normal spermatozoa were analyzed by t-test using the SPSS software at different levels of 0.05. The expression of the reproduction genes of both female and male tiger shrimp was descriptively presented.

Results

Isolation and identification of serotonin from the banana hump

The isolation result revealed that 14 g of serotonin was extracted from 2 kg of the dried banana hump, which was equivalent to 0.7%. The serotonin was characterized as orange-white powder, slightly soluble in water but easily soluble in alcohol, and a melting point of 167.7 °C. The HPLC chromatogram showed a sharp peak, indicating a very high purity of the serotonin compound (Fig. 1A) at a retention time of 2.0 min. The UV-Vis spectrum (Fig. 1B) showed absorption at a 205.0 nm wavelength, indicating an amide compound. The 221.1 and 277.0 nm peaks showed O-H and NH2 functional groups in the structure molecule.

Figure 1 HPLC chromatogram (A) and UV-Vis spectrum (B) of serotonin isolated from banana hump extract.

The IR spectrum of this compound (Fig. 2A) showed a broad spectrum at 3,402.43 cm−1 and two bands in the region 3,300–3,000 cm−1, namely at 2,958.80 and 2,927.94 cm−1 supporting the existence of an O-H, the asymmetrical N-H stretching, and the symmetrical N-H stretching group. A medium spectrum showed a C=C double bond at 1,666.50 cm−1. In addition, the presence of C-N bending and banding vibration of the aromatic group were assigned to the peak at 1,109.07 and 937.40–763.81 cm−1. The IR spectrum of this compound was similar to the IR spectrum of the serotonin standard (Fig. 2B).

Figure 2 The IR spectrum of serotonin isolated from banana hump extract (A) and a standard serotonin (B).

Tiger shrimp molting

The variation in the number of individuals molting every week after the injection of the serotonin extract is shown in Fig. 3. The number of shrimp molting per week for females (Fig. 3A), males (Fig. 3B), and pooled samples of females and males (Fig. 3C) exhibited a similar pattern. Up to 2-month observations, the number of individuals molting per week ranged from 1–4, with an average of 2.0 shrimp/week in the shrimp injected with serotonin extract and 1.6 shrimp/week in the control shrimp. The statistical analysis showed that the average number of broodstock molting per week did not show a significant difference (P > 0.05) between both treatments. However, a trend indication of the shrimp serotonin injection treatment was higher in number compared to the control treatment without herbal extract application. The present study revealed that the molting process occurred throughout the day, where the highest number of molting was obtained in the 2nd week after the herbal extract injection. The percentage of 100% molting or all individuals had molted in the banana hump extract treatment in the 5th week, and molting continued until the 8th week (2 months). Almost all tiger shrimp broodstock had molted twice, and the cumulative molting had reached 160% in the serotonin injection treatment and 140% in the control treatment (Fig. 3B).

Figure 3 The number of shrimp molting per week for female (A), male (B), pooled samples (C), and the cumulative percentage of shrimp molting for 8 weeks post-injection (D).

Treatment SE, serotonin extract of the banana hump; and CT, control treatment without application of serotonin extract.

Reproduction of female broodstock

The GML of the female tiger shrimp reproductive system is characterized by the growth of ovaries through the parallel differentiation and development of oocytes. Observations of the GML of tiger shrimp and broodstock after injection showed that the GML started to develop to GML-1 in the 1st week post-extract injection and reached GML-2 in the 2nd week post-injection. In the 3rd week after injection, three broodstocks at GML-3 were found in the serotonin extract treatment and one broodstock at GML-2 in the control treatment. The development of GML increased to GML-4 until spawning within 3 weeks after the herbal extract injection.

The application of banana hump extracts improved the reproductive tiger shrimp broodstock performance compared to controls (Table 1). The number of spawned tiger shrimp broodstock was obtained to be higher in the banana hump extract treatment than in the control treatment. Three out of five broodstocks (60%) spawned, while the control was still two broodstocks (40%). Even for the banana hump herbal extract treatment, it was found that one broodstock spawned twice or first re-maturation (20%) and three times or second re-maturation (20%). The total number of eggs from tiger shrimp broodstock was found to vary between 125,600–286,550 eggs/shrimp, where the first spawning (maturation) was higher in the banana hump extract treatment than in the control treatment, namely 125,500–171,670 eggs/shrimp. The statistical analysis between the treatments was significantly different (P < 0.05) in the number of eggs. There was also a trend for the number of eggs to be smaller in subsequent spawnings (re-maturation). The egg diameter between treatments did not show a significant difference in size (P > 0.05) with an average range of 274.6–275.4 µm, although the serotonin extract treatment showed a relatively higher egg diameter number.

Table 1 Number of female spawning, total number, and diameter of eggs of tiger shrimp broodstock post application of serotonin extract.

Treatment	Number of female broodstocks	Parameters	Maturation	The 1st re-maturation	The 2nd re-maturation	
SE	5	Number of spawning broodstock	3 (60%)	1 (20%)	1 (20%)	
Egg number	286,550 ± 46,402a	270,900	125,600	
Egg diameter (µm)	275.4 ± 8.4a	259.1 ± 5.7	285.1 ± 9.5	
CT	5	Number of spawning broodstock	2 (40%)			
Egg number	148,585 ± 23,647b			
Egg diameter (µm)	274.6 ± 6.1a			
Note:

SE, Serotonin extract of the banana hump; and CT, control treatment (without extract), and mean values followed by different superscript letters in the same parameter and column indicated a significant difference (P < 0.05).

Reproduction of male broodstock

The application of serotonin extracts showed greater male GML than without the application of herbal extracts. The development of GML in male broodstock injected serotonin extract showed a higher proportion of GML compared to the control treatment, namely four out of five shrimps (80%), consisting of one shrimp at GML-1, two shrimp at GML-2 and one shrimp at GML-3. In the controls, there were four out of five shrimps (80%), but only three of them reached GML-2, and one other shrimp was still at GML-1. Analysis of the number of spermatophores, spermatophore weight, number of spermatozoa, and percentage of normal spermatozoa of the GML-2 male tiger shrimp broodstock after extract injection was presented in Table 2.

Table 2 The number of spermatophore, spermatophore weight, number of spermatozoa, and normal spermatozoa in tiger shrimp after application of serotonin extract.

Parameters	SE	CT	
Number of spermatophore	2	2	
Spermatophore weight (g)	0.06 ± 0.01a	0.05 ± 0.01a	
Number of spermatozoa (106 cells)	33.31 ± 2.04a	12.03 ± 0.49b	
Normal spermatozoa (%)	36.55 ± 9.26a	20.80 ± 15.56a	
Note:

SE, Serotonin extract of the banana hump; and CT, control treatment (without extract), and mean values followed by different superscript letters in the same raw indicated significant differences (P < 0.05).

Two spermatophore sacs were observed in each broodstock, with no difference in spermatophore weight between tiger shrimp injected with serotonin extract and the control treatment. A total of 33.31 × 106 cells of spermatozoa was observed from broodstock-injected serotonin compared to the control of 12.03 × 106 cells, and statistical analysis showed that the broodstocks injected by serotonin were significantly higher (P < 0.05) compared with control groups. On the other hand, the percentage of normal spermatozoa was not significantly different (P > 0.05). However, a higher rate was observed in the serotonin application.

Expression of genes related to reproduction

After serotonin extract injection, the reproductive response was assessed by analyzing the expression of genes involved in reproduction, such as the vitellogenin gene in female tiger shrimp and the gametogenesis gene in males. The successful isolation of the genes was demonstrated by the clear and consistent band of β-actin gene expression, appearing at approximately 400 bp, in both female and male tiger shrimp. The gene expression analysis indicated that injecting serotonin extract enhanced reproductive quality. This was evidenced by a higher expression of the vitellogenin gene (around 200 bp) and the spermatogenesis gene (around 100 bp) than in the control group or those not treated with herbal extracts (Fig. 4).

Figure 4 Expression of the genes encoding reproduction of female and male tiger shrimp broodstock, M, DNA marker 50 bp; SE, serotonin extract of the banana hump; and CT, control treatment without application of serotonin extract; NC, negative control; arrow, position of gene target; 1, expression of β-actin gene in females; 2, expression of vitellogenin gene in females; 3, expression of β-actin gene in males; and 4, expression of gametogenesis gene in males.

Discussion

Serotonin is commonly known as 5-hydroxytryptamine, a type of hormonal compound that can be isolated from various plants. Based on the analyses of HPLC, UV-Vis, and IR spectrum, the serotonin was successfully isolated and characterized from the banana hump. The findings of this study demonstrated that the banana hump was a good source of serotonin compounds with a relatively high content. Serotonin was also present in many plant parts (stems, leaves, and flowers), and it was involved in the physiology and regulation of growth, xylem sap exudation, flowering, ion permeability, and plant morphogenesis within the plant system (Ramakrishna, Giridhar & Ravishankar, 2011). For this study, serotonin was assessed to determine its effect on the tiger shrimp broodstock’s molting, reproduction performance, and gene expression.

Molting is a process of shedding their crustacean exoskeleton, often called ecdysis, which is an essential event in crustacean physiology because the molting cycle is linked to maturation (Lemos & Weissman, 2021). The higher number of tiger shrimp molting in the herbal extract treatment indicated that the active serotonin compound from banana hump played a function in inducing the molting process in tiger shrimp, especially in inducing broodstock reproduction. Furthermore, the molting process plays an essential role in the shrimp reproductive cycle; during molting, female shrimp can produce eggs, and male shrimp can produce sperm, so the molting allows them to prepare themselves for the reproductive process. The molting can be influenced by stress or environmental changes, and hormonal compounds, including ecdysterone and serotonin compounds, can control skin turnover in shrimp (Promwikorn, Kirirat & Thaweethamsewee, 2004).

Injecting herbal extracts such as serotonin from banana plants can stimulate and increase the molting process in tiger shrimp. A higher frequency of cumulative molting in tiger shrimp broodstock injected with herbal extract compared to those without injection has been reported by several previous studies (Rosmiati, Lante & Suryati, 2016; Suryati, Tenriulo & Tonnek, 2013; Suryati et al., 2021). The occurrence of molting can be influenced by several aspects, including stress or environmental changes. Hormonal compounds include ecdysterone and serotonin, which can control shrimp molting (Promwikorn, Kirirat & Thaweethamsewee, 2004). Uddin & Rahman (2016) reported that all ablated females of tiger shrimp broodstocks were molted within 7 days, and the first spawning occurred within 3 days after eyestalk ablation.

This study demonstrated that injecting serotonin extract resulted in more broodstock and eggs than the control group. Specifically, three female broodstocks in the serotonin extract group spawned with an average fecundity of 286,550 eggs. In contrast, only two females in the control group spawned, producing an average fecundity of 163,585 eggs over the 2-month investigation period. The fecundity of the broodstocks correlated with their body weight. A significant positive correlation (r = 0.82) between tiger shrimp fecundity and female body weight was observed. The study also revealed that female re-maturation occurred with serotonin extract injection, with a 20% re-maturation rate for both the first and second maturation. The significant number of spawned females and re-maturation occurrence suggested that serotonin extract effectively stimulated gonad development and enhanced the reproduction of tiger shrimp. A similar finding regarding the egg diameter of spawned tiger shrimp was reported in some studies. The ablated female produced an average of 296,160 ± 26,589 eggs, while a non-ablated female produced 195,462 ± 20,565 eggs during 120 days of the observation period (Uddin & Rahman, 2016). They also reported that among the ablated tiger shrimp females, 36% spawn once (maturation), 48% the second time (the 1st re-maturation), and 16% the third time (the 2nd re-maturation).

Serotonin has been reported as a reproductive stimulant in crustacean species, aiding in the induction of ovarian maturation and spawning in the crayfish Procambarus clarkii (Sarojini, Nagabhushanam & Fingerman, 1995) and the white Pacific shrimp Penaeus vannamei (Vaca & Alfaro, 2000). According to Wongprasert et al. (2006), the impact of administering exogenous serotonin at a dose of 50 μg/g body weight on the reproduction of black tiger shrimp demonstrated ovarian maturation and spawning rates similar to those achieved through ablation. Their finding also showed that the hatching rate and nauplii number per spawning were significantly higher in serotonin-injected shrimp compared to eye-stalk-ablated shrimp. Moreover, the serotonin-positive reaction was obtained in pre-vitelogenic oocyte follicular cells, early vitelogenic oocytes’ cytoplasm, and the cell membrane and cytoplasm of late vitelogenic oocytes. The results suggest an essential role for serotonin, possibly directly in the ovaries and oviducts, in the reproductive function of female tiger shrimp (Wongprasert et al., 2006). The findings of this present study indicated that female tiger shrimp broodstock was induced to stimulate ovarian maturation and spawning by injecting 50 μg/g body weight of serotonin extract. According to Sarojini, Nagabhushanam & Fingerman (1995) and Vaca & Alfaro (2000), the serotonin injection at the same dose used in Penaeus monodon and Penaeus vannamei also effectively induces ovarian maturation. A similar result was reported in the freshwater prawn Macrobrachium resenbergii, whose ovarian index was significantly enhanced by serotonin injection at a lower dose of 1 μg/g body weight (Meeratana et al., 2006).

Even though the spermatophore weight did not significantly differ between both treatments, the trend of higher spermatophore weight was obtained in the serotonin-injected shrimp. Compared with the previous study, a spermatophore weight of 0.041 g was reported in male captive tiger shrimp broodstock post-injection with 17α-methyltestosterone (Rosmiati et al., 2022). Based on the results of this present study, the reproduction of male shrimp could be induced by serotonin injection, which may also enhance the gene expression related to gametogenesis in male broodstock (see Fig. 4). The result showed that the spermatozoa number of male tiger shrimp broodstock was significantly higher than the control group.

It is suspected that the active serotonin compound induced directly and indirectly the reproductive development of female and male tiger shrimp broodstocks. The reproductive gene expression analysis has been equipped with internal control in the form of tiger shrimp β-actin gene expression. However, reproductive improvement of tiger shrimp broodstock through quantitative gene expression analysis still needs to be done, for example, by gene expression analysis using real-time q-PCR. The increase in gene expression related to gametogenesis in male broodstock is influenced by herbal extract stimulants introduced from outside. The results of this study indicated that serotonin extract compounds increased gene expression of gametogenesis.

The observed increase in fecundity, particularly in the number of eggs produced per spawning, can be attributed to the upregulation of the vitellogenin gene in serotonin-treated shrimp. The result indicated a direct role in enhancing oocyte development and maturation, which is reflected in the larger number of eggs observed in this study. Vitellogenin plays a key role in yolk production, which is essential for egg quality and embryo development (Chen et al., 2022; Tomy et al., 2016). The higher fecundity in the serotonin-treated broodstock (286,550 eggs vs. 148,585 eggs in the control) likely resulted from the enhanced vitellogenesis process, as indicated by the elevated gene expression levels. This suggested that serotonin not only influences the quantity of eggs produced but also enhances their developmental potential by improving vitellogenesis. Similarly, the increased expression of the spermatogenesis gene in male broodstock indicates that serotonin stimulates the production of sperm cells, leading to a higher sperm count and improved reproductive success (Fujinoki, 2011). A higher number of total spermatozoa was observed in the serotonin group (33.31 × 106 cells) than in the control group (12.03 × 106 cells) further supported the conclusion that serotonin positively influences reproductive health in Penaeus monodon. It is also known that gametogenesis is a process that produces spermatogenesis (formation of male sex cells) or oogenesis (formation of female sex cells) (Wijayanti, Soeminto & Simanjuntak, 2009). Increased reproduction performance of female tiger shrimp given methyl fernesoate extract from the grass herb has been reported using a molecular analysis approach through the vitellogenin gene expression (Suryati et al., 2021). The results of this present research provided an important insight into reproductive tiger shrimp broodstock for supporting tiger shrimp domestication and producing offspring with good quality both genotypically and phenotypically.

The application of serotonin from banana hump offers promising opportunities for sustainable broodstock management in commercial aquaculture systems. This study indicated that serotonin from the banana hump has a potential reproductive stimulant for tiger shrimp broodstock by influencing the molting frequency and reproductive processes in shrimp by acting as plant-derived neurohormonal pathways. By addressing these practical and scalable applications, this study not only advances the scientific understanding of serotonin’s role in shrimp reproduction but also provides actionable insights for improving broodstock management in commercial aquaculture. Future studies should explore the scalability of serotonin extraction and application in larger aquaculture operations, as well as its potential effects on other economically important marine species. Even though the serotonin in this study significantly enhanced the reproduction performance of tiger shrimp broodstock, for eradicating eyestalk ablation, the recommended further research on the treatment of injected vs non-ablated tiger shrimp broodstock and the development of serotonin application through oral administration.

Conclusions

The active compound, serotonin, was successfully isolated from banana humps and identified using HPLC, UV-Vis, and IR spectroscopy analyses. This serotonin compound accelerated molting and enhanced reproductive performance in female and male tiger shrimp broodstock. Shrimp injected with the serotonin extract showed higher expression of reproduction-related genes than the control group. Thus, injecting serotonin at a dose of 50 µg per body weight could be an effective stimulant for boosting reproductive performance in tiger shrimp broodstock.

Supplemental Information

Supplemental Information 1 The average of molting shrimp broodstock, total number of eggs, egg diameter, and the number of spermatozoa from shrimp broodstock treated with serotonin (SE) and from untreated shrimp broodstock.

Higher values indicate better reproductive performance of the tiger shrimp broodstock.

The data were statistically analyzed to identify the effect of serotonin administration on the reproductive performance of the black tiger shrimp broodstock.

The authors thank the technicians of the Biotechnology Laboratory of BRIN and RIBAFE, and all those who provided their help throughout the study. The authors further acknowledge the use of ChatGPT to assist in partially writing and editing this manuscript.

Additional Information and Declarations

Competing Interests

Author Contributions

Data Availability

The authors declare that they have no competing interests.

Andi Parenrengi conceived and designed the experiments, performed the experiments, analyzed the data, prepared figures and/or tables, authored or reviewed drafts of the article, and approved the final draft.

Emma Suryati conceived and designed the experiments, performed the experiments, analyzed the data, prepared figures and/or tables, authored or reviewed drafts of the article, and approved the final draft.

Rachman Syah conceived and designed the experiments, performed the experiments, prepared figures and/or tables, authored or reviewed drafts of the article, and approved the final draft.

Andi Tenriulo performed the experiments, prepared figures and/or tables, authored or reviewed drafts of the article, and approved the final draft.

Samuel Lante performed the experiments, prepared figures and/or tables, authored or reviewed drafts of the article, and approved the final draft.

Elmi Nurhaidah Zainuddin performed the experiments, analyzed the data, prepared figures and/or tables, authored or reviewed drafts of the article, and approved the final draft.

Ratu Siti Aliah performed the experiments, analyzed the data, authored or reviewed drafts of the article, and approved the final draft.

Nuril Farizah performed the experiments, analyzed the data, authored or reviewed drafts of the article, and approved the final draft.

Agus Nawang performed the experiments, authored or reviewed drafts of the article, and approved the final draft.

Sulaeman Sulaeman performed the experiments, authored or reviewed drafts of the article, and approved the final draft.

Makmur Makmur performed the experiments, authored or reviewed drafts of the article, and approved the final draft.

Rosmiati Rosmiati performed the experiments, analyzed the data, prepared figures and/or tables, authored or reviewed drafts of the article, and approved the final draft.

Gunarto Gunarto performed the experiments, authored or reviewed drafts of the article, and approved the final draft.

Herlinah Herlinah performed the experiments, authored or reviewed drafts of the article, and approved the final draft.

The following information was supplied regarding data availability:

The raw measurements are available in the Supplemental File.

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
