# Peer review of "Isolation and identification of serotonin compound from banana hump: a reproductive stimulant for tiger shrimp Penaeus monodon broodstock enhancement"

_PeerJ, doi:10.7717/peerj.18670_

## Round 0.1 · original submission · Major Revisions

The manuscript has been reviewed, and while the reviewers agree the study is essential, some significant issues need fixing. The authors are required to fix the considerable grammar issues with the manuscript. The material and methods need more detail on how the procedures were performed. Please address the concerns raised by the reviewers.

Reviewer 1 ·

Basic reporting

### 1. **Clear and Unambiguous, Professional English Used Throughout**
The manuscript is generally well written, with a clear scientific tone and the use of professional language. However, there are areas where the clarity of sentences could be improved, as well as some minor grammatical issues. Here are specific recommendations:
- Improve clarity in the description of the extraction method, especially in lines regarding the volumes and weights (e.g., rephrase "500 g added to water until 2 L" to ensure accuracy).
- Some terms and explanations could benefit from more precise language. For example, in the methods, there is ambiguity regarding certain procedures (e.g., centrifugation steps and the use of specific temperatures for extraction). Make these descriptions more straightforward to aid reproducibility.

**Suggested Improvement:**
- I recommend a language review by a profficient English speaker or a scientific editor familiar with aquaculture terminology. Additionally, clarify steps in the methodology, especially related to the extraction and gene expression procedures.

### 2. **Literature References, Sufficient Field Background/Context Provided**
The manuscript provides a reasonable background on serotonin's role in shrimp reproduction and includes several relevant citations. However, some referenced studies appear tangential to the specific serotonin-related research focus of this study. For instance, references discussing other compounds like methyl fernesoate, phytoecdysteroids, and RNAi, while relevant to shrimp reproduction, detract from the focus on serotonin.

**Suggested Improvement:**
- To maintain focus, the authors should streamline the discussion to emphasize serotonin-related studies more prominently. Literature concerning serotonin's application in aquaculture and its physiological mechanisms in invertebrates should be expanded.
- The discussion of other non-serotonin compounds, such as methyl fernesoate and testosterone, should be minimized unless directly comparing physiological mechanisms.

### 3. **Professional Article Structure, Figures, Tables. Raw Data Shared**
The structure of the article follows the standard format, but several sections, such as the methods and results, require more details to ensure replicability. Figures are relevant and appropriately labeled, but there could be improvements in resolution and clarity, especially with regard to the gene expression data presented in Figure 4.

**Suggested Improvement:**
- Ensure all figures, especially chromatograms and gene expression images, are of the highest possible resolution and clarity.
- Clarify the methods for HPLC and gene expression analysis, as some critical details (e.g., temperatures for various steps and concentration details for the reagents) are vague.

### 4. **Self-contained with Relevant Results to Hypotheses**
The submission is self-contained and presents clear hypotheses about the role of serotonin extracted from banana humps in enhancing tiger shrimp reproduction. However, the connection between gene expression findings and reproductive outcomes (e.g., egg number and diameter) is not fully explained. Specifically, the link between vitellogenin expression and egg diameter needs more detailed discussion.

**Suggested Improvement:**
- The authors should expand the discussion to explicitly connect the gene expression data (vitellogenin and spermatogenesis genes) with the observed reproductive outcomes. How does increased gene expression translate into practical reproductive performance metrics (egg diameter, fecundity)?

### Overall Comment:
This is a promising study on the potential role of serotonin from banana hump in tiger shrimp broodstock enhancement. It addresses an important gap in sustainable aquaculture practices. With revisions to the language, methodology details, and some restructuring of the discussion to focus on serotonin-related literature, this manuscript will be significantly strengthened.

Experimental design

**1. Original primary research within Aims and Scope of the journal**

The research question is well defined and highly relevant to the field of aquaculture, specifically in improving reproductive outcomes for tiger shrimp broodstock using natural compounds. The study addresses a significant knowledge gap in the use of serotonin extracted from banana hump as an alternative to traditional reproductive stimulants such as eyestalk ablation. This aligns well with the journal's aims of publishing innovative and impactful research on aquaculture practices.

**Suggested Improvement:**
- While the research gap is identified, the introduction could be improved by clearly articulating how this specific study builds on previous work. Emphasize why serotonin, specifically from banana hump, is a more sustainable or effective option compared to other methods already established in the field.

**2. Rigorous investigation performed to a high technical & ethical standard**

The study appears to be conducted to a high technical standard, and ethical approval has been obtained from the appropriate institutional body. The experiments are logical, and the ethical guidelines for handling broodstock have been followed.

**Suggested Improvement:**
- The study could benefit from a more detailed description of the ethical considerations, especially around the repeated injections and the use of serotonin in tiger shrimp. Mentioning any welfare monitoring of the shrimp throughout the experiment would further assure the reader of compliance with ethical standards.

**3. Methods described with sufficient detail & information to replicate**

The methods section provides a good overview of the experimental procedures. However, several critical steps lack sufficient detail for full reproducibility, particularly in the description of the extraction process and the gene expression analysis.

**Suggested Improvements:**
- Include more precise information on the volumes, temperatures, and equipment used during the serotonin extraction process. For example, specific centrifugation speeds and times, as well as precise temperature settings, should be provided.
- In the gene expression section, additional details about the controls used in the PCR analysis would be helpful, including how primers were validated and any specific steps taken to ensure reproducibility across samples.

Validity of the findings

**1. Impact and novelty not assessed. Meaningful replication encouraged where rationale & benefit to literature is clearly stated**

The study presents new findings on the potential use of serotonin extracted from banana hump as a reproductive stimulant in tiger shrimp, which offers a novel approach to improving shrimp broodstock performance. The rationale for the study is clear, and it addresses a specific gap in aquaculture research. However, the manuscript could provide a stronger rationale for the novelty and the potential impact of these findings beyond the immediate scope of shrimp reproduction.

**Suggested Improvement:**
- Expand on how this research contributes to broader aquaculture practices and sustainability. Consider discussing the potential applications of this serotonin extract in other species or broader commercial settings to enhance the perceived impact and relevance of the study.
- Clarify whether this study is replicating or building upon previous serotonin-based research, and if so, provide more justification for the replication or comparison.

**2. All underlying data have been provided; they are robust, statistically sound, & controlled**

The data presented appears robust, with statistical analysis appropriately applied. However, some aspects of the data handling and analysis could be clarified to ensure the results are fully understood and reproducible.

**Suggested Improvement:**
- Provide more information about the data analysis methods, particularly in the statistical tests used to compare treatment groups. For example, clarify how normality and homogeneity of variance were tested, and whether any adjustments were made for multiple comparisons.
- Ensure that all raw data is made available either within the manuscript or through a data repository, particularly the gene expression data.

**3. Conclusions are well stated, linked to original research question & limited to supporting results**

The conclusions are generally well aligned with the data presented, focusing on the impact of serotonin extract on reproductive performance in shrimp. However, some claims regarding the broader applicability and effectiveness of serotonin could be more conservative, as they rely on indirect evidence.

**Suggested Improvement:**
- Ensure that conclusions about the efficacy of serotonin as a reproductive stimulant are based solely on the results of this study. Avoid overgeneralizing the findings to other species or systems unless supported by additional evidence.
- The discussion of gene expression results could benefit from more cautious interpretation, particularly when linking these results to observed phenotypic changes like egg diameter.

Additional comments

Overall, the study presents interesting and valuable findings on the potential use of serotonin from banana hump as a reproductive stimulant in tiger shrimp, with implications for improving broodstock management in aquaculture. The research is relevant, and the methodology is sound. However, a few areas can be strengthened:

- The manuscript would benefit from a clearer explanation of the physiological mechanisms by which serotonin influences reproduction and molting in shrimp. A more detailed discussion of the serotonin pathways involved could deepen the readers' understanding.

- The integration of literature could be more focused, with an emphasis on serotonin-related studies to avoid distraction by unrelated compounds like methyl fernesoate or ecdysteroids. This will help maintain the manuscript's focus on serotonin as a novel reproductive stimulant.

- Consider adding more practical implications or future directions for the application of this research in broader commercial aquaculture systems. This could help readers appreciate the potential real-world impact of your findings.

Overall, with these revisions, this manuscript will contribute significantly to the field of sustainable aquaculture and shrimp reproductive enhancement.

Annotated reviews are not available for download in order to protect the identity of reviewers who chose to remain anonymous.

·

Basic reporting

My review is included as attachment.

Experimental design

Comments in attached document

Validity of the findings

Comments in attached document

---

## Round 0.2 · Minor Revisions

Could you please comment on the minor issue raised reviewer 2

Reviewer 1 ·

Basic reporting

No comment

Experimental design

No comment

Validity of the findings

No comment

Additional comments

The manuscript has been revised according to the previous suggestions and shows significant improvement. These changes have enhanced the overall structure, clarity in methodology, and data interpretation. Only minor adjustments are needed to reach the final optimal standard for acceptance.

Annotated reviews are not available for download in order to protect the identity of reviewers who chose to remain anonymous.

·

Basic reporting

The flow of the manuscript has been improved.
The results and interpretations are now clearer. Sufficient references have been provided for background and discussion. Figures and tables were now provided for revision and are of good quality

Experimental design

The experimental design has been improved with more details about specific processes.
The scope of the manuscript has been better explained and better fits the aims and scope of originality of the journal.

Validity of the findings

Authors re-stated that the goal of the manuscript was not to seek alternatives to welfare but to isolate and identify the serotonin compound of the banana hump, as well as to evaluate its effect on the molting rate, reproduction, and gene expression of tiger shrimp broodstock, compared to the control shrimp without serotonin extract herb injection. Introduction in lines 95-98. The conclusions are consistent with the aims of the manuscript.

Raw data has been provided.

Additional comments

Sufficient references have been provided for background and discussion. However, in LINE 163 I noted that reference Parenrengi et al. (2021) has not been provided at the end of document. This reference is important as it would help clarifying how authors synthetised cDNA from haemolymph. Did authors obtained RNA from hemocytes recovered from haemolymph? or hemocytes were lysed in haemolymph and RNA was recovered as fragments? Could authors explain this and provide the appropriate referencing (Parenrengi et al. (2021)?

---

## Round 0.3 · accepted · Accept

Authors have addressed all of the reviews comments.